# Instruments for Assessing Nursing Care Quality: A Scoping Review

**DOI:** 10.3390/nursrep15090342

**Published:** 2025-09-19

**Authors:** Patrícia Correia, Rafael A. Bernardes, Sílvia Caldeira

**Affiliations:** 1Unidade Local de Saúde de Santa Maria (ULSSM), Hospital de Santa Maria, 1649-035 Lisbon, Portugal; 2Centre for Interdisciplinary Research in Health (CIIS), Faculty of Health Sciences and Nursing (FCSE), Universidade Católica Portuguesa (UCP), 4169-005 Lisbon, Portugal; rbernardes@ucp.pt (R.A.B.); scaldeira@ucp.pt (S.C.)

**Keywords:** instruments, quality, nursing care, scales

## Abstract

**Background/Objectives.** Quality of nursing care (QNC) is a central concept in healthcare systems worldwide, with growing emphasis on developing reliable and contextually appropriate instruments for its assessment. Over recent decades, there has been a shift from outcome-based evaluation toward more holistic, patient-centered frameworks that consider both clinical indicators and interpersonal dimensions of care. This scoping review aimed to map the range, nature, and characteristics of self-report instruments used to assess the quality of nursing care, including their psychometric properties and contextual applications across different clinical settings. **Methods.** A systematic search was conducted in CINAHL Complete, MEDLINE (via PubMed), Scopus, Web of Science, and ProQuest Dissertations & Theses, alongside gray literature sources, following the Joanna Briggs Institute (JBI) methodology and PRISMA-ScR guidelines. Studies were included if they reported on the development, validation, adaptation, or application of QNC assessment tools in hospital or community nursing contexts, and were published in English, Portuguese, or Spanish. **Results.** Fifty-nine studies were included, spanning from 1995 to 2025. The instruments identified were predominantly structured around Donabedian’s structure-process-outcome model, and many emphasized relational domains such as empathy, communication, and respect. Tools like the Good Nursing Care Scale (GNCS), the Quality of Oncology Nursing Care Scale (QONCS), and the Karen Scales demonstrated strong internal consistency (Cronbach’s α ranging from 0.79 to 0.95). **Conclusions.** Organizational factors, including leadership and staffing, and predictors such as burnout and work intensity, were found to influence perceived care quality. Important gaps remain regarding longitudinal use and integration of patient-reported outcome measures.

## 1. Introduction

The quality of nursing care (QNC) is critical to healthcare delivery, directly influencing patient outcomes, safety, satisfaction, and system efficiency [1]. As healthcare systems evolve toward more patient-centered, evidence-based, and outcome-driven models, the need for robust, multidimensional, and context-sensitive tools to assess QNC becomes increasingly urgent [2]. Despite the proliferation of instruments designed to measure QNC, considerable heterogeneity persists in their conceptual foundations, target populations, and psychometric robustness [3]. Traditionally, evaluations of nursing care quality focused primarily on patient outcomes or institutional performance indicators. However, there is increasing recognition of the importance of incorporating the perspectives of nurses—those directly involved in care delivery [4]. Nurses’ insights provide a nuanced understanding of care processes, organizational dynamics, and contextual challenges that may not be captured through patient or administrative data alone [5].

Several conceptual frameworks have guided the development of QNC assessment tools. Donabedian’s model, which categorizes quality into structure, process, and outcomes, remains a foundational reference in healthcare quality evaluation [6]. More contemporary approaches, such as evidence-based practice (EBP), emphasize the integration of clinical expertise, patient preferences, and the best available evidence to guide care decisions [7].

To complement these perspectives, Watson’s Theory of Human Care provides an essential framework for understanding quality as more than technical competence. Watson emphasizes the importance of caring relationships, empathy, communication, and the holistic integration of physical, emotional, and spiritual needs in the healing process.

While the systematic review by Koy and colleagues [8] offers a valuable foundation by listing quantitative instruments for measuring nursing care quality, several distinctions justify the need for this scoping review. Their review was limited to quantitative studies published in English up to 2015, whereas the present study adopts broader inclusion criteria—encompassing qualitative, mixed-methods, and gray literature—to provide a more comprehensive and current synthesis. Furthermore, this review identifies emerging themes such as nurse well-being, culturally adapted tools, and the integration of patient-reported outcomes, offering practical insights for diverse healthcare contexts.

The aim of this review is to map existing QNC measurement tools, identify conceptual and methodological gaps, and provide a comprehensive synthesis of how quality is defined, operationalized, and validated across healthcare settings. Mapping this evidence may support stakeholders in selecting or developing appropriate instruments tailored to their specific contexts and goals. Importantly, this review acknowledges that quality in nursing care is a multidimensional phenomenon that combines technical competence, interpersonal relationships, organizational support, and patient engagement. In doing so, it aligns both with Donabedian’s structural-process-outcome framework [6] and with Watson’s emphasis on human caring, empathy, and holistic patient needs, reflecting contemporary models of co-produced healthcare quality [2,9].

## 2. Materials and Methods

### 2.1. Study Design

This scoping review was conducted following the methodology proposed by the Joanna Briggs Institute (JBI) and is reported according to the Preferred Reporting Items for Systematic Reviews and Meta-Analyses extension for Scoping Reviews (PRISMA-ScR) [10]. The objective was to map the available evidence regarding self-report instruments used to assess the quality of nursing care on self-reported instruments used to assess the quality of nursing care, with particular focus on their psychometric properties, underlying conceptual structures, and clinical application contexts. A preliminary search of PROSPERO, MEDLINE, the Cochrane Library, and JBI Evidence Synthesis databases revealed no existing or ongoing scoping or systematic reviews on this specific topic at the time of the search.

The review protocol was registered in OSF (https://doi.org/10.17605/osf.io/XUVHW).

### 2.2. Review Question

This scoping review aimed to systematically map and synthesize the literature on instruments used to assess the quality of nursing care (QNC).

Specific questions were:Clinical contexts: in which healthcare settings (e.g., hospital wards, community care, specialized units) have QNC assessment instruments been developed, adapted, or implemented?Instrument characteristics: what are the structural, conceptual, and methodological features of these instruments, including format, length, target respondents, and theoretical concepts?Domains of nursing care quality: which dimensions of nursing care quality (e.g., technical competence, interpersonal relationships, patient-centeredness, holistic care) are captured by these instruments?Psychometric properties: what evidence exists regarding the reliability, validity, and overall measurement robustness of these instruments, including cross-cultural adaptations and longitudinal evaluations?

### 2.3. Eligibility Criteria

#### 2.3.1. Participants

Studies were eligible if they involved nurses of any training level, clinical specialty, or practice setting. Both staff nurses and nurse managers were considered if the focus was on the nurses’ perspective in evaluating quality of care.

#### 2.3.2. Concept

The central concept was the assessment of nursing care quality using self-report instruments, including questionnaires, rating scales, checklists, or other standardized tools. Only instruments explicitly designed to measure the quality of care were included. Tools that exclusively evaluated individual nursing competencies, technical skills, or task performance without addressing broader care quality were excluded. The perspective of the nurse as care provider was central; studies capturing patient or organizational perspectives without a nurse-reported component were not included.

#### 2.3.3. Context

Eligible studies were conducted in any healthcare setting where nursing care is delivered, including hospital units (e.g., intensive care, perioperative, oncology, palliative care), outpatient services, and community-based care. No restrictions were applied based on patient population, clinical specialty, or geographic location.

#### 2.3.4. Types of Sources

This review included:Primary empirical studies, both quantitative and qualitative;Studies focused on instrument development or validation;Studies involving translation and cross-cultural adaptation instruments;Gray literature, such as dissertations.

Inclusion Criteria:


Studies that describe, validate, apply, or adapt instruments to assess the quality of nursing care.Publications available in English, Portuguese, or Spanish.Studies providing sufficient methodological detail to evaluate the instrument and its domains.


Exclusion Criteria:


Instruments focused solely on individual nursing competencies (e.g., technical skills or proficiency) without addressing broader aspects of care quality.Theoretical works lacking empirical testing or practical application.Studies where nurses were not a primary respondent group for the instrument.


### 2.4. Search Strategy

The search strategy was computed in the following databases: CINAHL Complete, PubMed, Scopus, Web of Science and ProQuest. The detailed queries can be found in Appendix A.

### 2.5. Study Selection

All identified records from the database and gray literature searches were imported into Rayyan for deduplication. Following removal of duplicates, the selection process was carried out in three stages: (1) Title Screening—Two reviewers independently screened all titles to exclude clearly irrelevant records. At this stage, studies obviously outside the scope—such as those not involving nurses, not related to quality of care, or purely theoretical works—were excluded; (2) Abstract Screening—Abstracts of remaining records were independently assessed by the same reviewers. Studies were included if they appeared to meet the eligibility criteria for participants, concept, context, and study type. Ambiguous abstracts were retained for full-text review to avoid premature exclusion; and (3) Full-Text Screening—Full texts of potentially eligible studies were retrieved and independently evaluated by the two reviewers. Each study was assessed against the inclusion and exclusion criteria.

Disagreements at any stage were resolved through discussion. If consensus could not be reached, a third reviewer was consulted.

### 2.6. Data Extracion

Data was extracted using a standardized form developed by the reviewers. Extracted data included:Authors, year, and country;Study objective;Name and type of instrument;Conceptual definition of quality;Clinical context of application;Domains and dimensions assessed;Psychometric properties (e.g., Cronbach’s alpha, construct validity, test–retest reliability);Summary of main findings.

The data extraction tool was updated iteratively as needed throughout the extraction process. Any disagreements between reviewers were resolved through discussion or, when necessary, by consultation with a third reviewer. In cases where key data were missing, attempts were made to contact the original study authors for clarification.

### 2.7. Data Analysis and Presentation

Extracted data were synthesized descriptively and presented in both tabular and narrative formats, organized around the review questions. The narrative synthesis summarized the types of instruments, evaluated domains, clinical contexts, and psychometric characteristics.

## 3. Results

### 3.1. Overall Characterization of the Studies

This scoping review included 45 studies (Figure 1).

The overall research was conducted in 24 distinct countries or regions (Table 1). Countries with the most frequent contributions include Turkey (5 studies), China (4 studies), and Jordan (3 studies), Sweden (3 studies), and Portugal (3 studies). The preponderance of studies (approximately 88%) is quantitative, primarily utilizing surveys, instrument validation, or cross-sectional designs. A smaller proportion includes qualitative or phenomenological studies (4 studies), a modified Delphi process (1 study), and systematic reviews (2 studies). Research settings are varied, predominantly focusing on general and multi-hospital environments, alongside numerous studies in university/teaching hospitals and specialized units such as Intensive Care Units (ICUs), psychiatric wards, emergency departments, and palliative care centers. Sample sizes are highly variable, with patient samples ranging from 30 to 5536 individuals, and nurse/nursing staff samples spanning from 10 to 3451 professionals, also including head nurses and managers.

The total sample of nurses exceeds approximately 15,000 participants. This number includes large multicenter surveys involving thousands of nurses, such as 3451 nurses in Portugal, 1201 in Belgium, 744 in Fiji, 784 in China, 394 in Thailand, 310 in Iran, and 347 in Turkey.

### 3.2. Definition of Quality of Care and Related Factors

Across the reviewed studies, quality of nursing care is consistently conceptualized as a multidimensional construct that integrates technical, interpersonal, and organizational dimensions. Donabedian’s Structure–Process–Outcome (SPO) model [6] remains the predominant theoretical framework underpinning several instruments, including the Karen Scale, Good Nursing Care Scale (GNCS), CNCQS, and QONCS [54]. Within this framework, quality is defined through the dynamic interplay of structural factors (e.g., staffing levels, workload, and physical environment), nursing processes (assessment, planning, delivery, evaluation), and outcomes, such as patient satisfaction and health improvements [14,45,50,53].

A comparative synthesis of the instruments highlights four recurring dimensions that consistently define quality nursing care across contexts:1.Empathy and Patient-Centeredness, particularly emphasized in oncology and palliative instruments (e.g., QONCS, PNCQS) [9,24,42,49,50];2.Communication and Interpersonal Relationships, including effective dialog, teamwork, and collaboration (e.g., GNCS, Karen Scale) [20,48,54];3.Leadership and Professional Competence, referring to both individual expertise and the organizational support needed for safe, coordinated care (e.g., Monitor, SERVQUAL-based tools) [16,22,29,35];4.Work Environment and Structural Context, covering staffing adequacy, workload, fatigue, and organizational culture (e.g., SERVQUAL, studies of burnout and staffing adequacy) [14,50,52,53].

Holistic care—integrating emotional, spiritual, and social support—emerges as a unifying theme, especially in palliative and oncology contexts [24,49,50]. In these studies, nursing quality is framed as extending beyond technical execution to encompass empathetic, respectful, and intentional interactions that also address family involvement. Additional dimensions such as autonomy, documentation, and professional responsibility are also frequently reported as core components of nursing care quality [2,24,50,52].

Some instruments conceptualize quality explicitly as the gap between patient expectations and delivered care, emphasizing domains like reliability, responsiveness, assurance, and empathy [53]. From this perspective, nursing care quality is shaped not only by technical proficiency but also by the capacity to meet human needs through advocacy and caring, mediated by contextual factors such as staffing adequacy, workload management, and empowerment [50,52].

In specialized fields such as psychiatric and palliative nursing, quality indicators expand to include psychosocial relationships, ethical responsibility, patient safety, job satisfaction, and openness [33,53]. Collectively, these findings demonstrate that nursing care quality is a complex, multidimensional construct: it is measurable through validated instruments, but also subjectively experienced, encompassing empathy, communication, leadership, and environment as recurring pillars across cultural and clinical contexts worldwide [50,52,54].

### 3.3. Context of Studies

The majority of the studies were conducted in acute care hospital settings, with an increasing number focusing on specialized areas such as oncology, palliative care, and psychiatric services. For instance, oncology-focused research includes works by Charalambous and Adamakidou [24] and Marcomini et al. [49], while palliative care has been explored in studies like Zulueta Egea et al. [36], and psychiatric nursing examined by Alsyouf et al. [33] and Moen et al. [38]. Some studies concentrated on national or cross-cultural adaptations of established instruments, such as the Cambodian Nursing Care Quality Scale (CNCQS) adapted for Iran, Cambodia, and Mongolia, while others developed tools tailored to specific contexts, like the EPAECQC in Portugal and the Quality Nursing Care Scale—Turkish version (QNCS-T).

These studies encompass a broad spectrum of healthcare settings and sample sizes, reflecting diverse nursing environments worldwide. Hospital-based research predominates, including medical-surgical wards, elderly care units, specialized departments like oncology, intensive care units (ICUs), and emergency departments. Sample sizes vary widely, from small qualitative studies with around a dozen participants to large multicenter quantitative investigations involving thousands of nurses and patients.

Many studies took place in university-affiliated or teaching hospitals, often engaging hundreds of nurses and patients, such as those conducted in China, Turkey, and Italy. Multi-hospital studies are common, spanning multiple institutions within countries or regions and including general hospitals, tertiary care centers, and specialty units. Notable examples include over 3400 nurses across public hospitals in Portugal, more than 1200 nurses in various hospital settings in Belgium, and hundreds of nurses participating in hospital networks in Korea, Mongolia, and Turkey.

Smaller, focused investigations often targeted specific units like ICUs or palliative care, with sample sizes ranging from several dozen to a few hundred nurses. Although less frequent, community and mental health settings were also studied, involving specialized samples of mental health professionals. Qualitative and expert panel studies, while involving fewer participants, provided valuable in-depth insights through interviews and Delphi techniques with nursing experts, clinicians, and educators.

### 3.4. Psychometric Properties

The measurement of nursing care quality across numerous studies reflects robust psychometric properties. Reliability evidence consistently indicates strong internal consistency, with Cronbach’s alpha values generally ranging from approximately 0.65 to 0.97. Many instruments report alpha coefficients exceeding 0.90, signifying excellent reliability [50,54]. Test–retest reliability, another critical indicator of stability over time, shows strong correlations often between 0.79 and 0.96 [50]. Inter-rater reliability, where applicable, is also high, with Intraclass Correlation Coefficients (ICC) spanning from 0.60 to 0.98, demonstrating consistent scoring across different evaluators [54].

Split-half reliability and person-separation indices further affirm the internal consistency and reliability of nursing care quality scales across different settings [52]. Instruments have undergone item refinement, often reducing lengthy original scales—for example, from 74 to 35 items—without sacrificing psychometric robustness, thus improving practicality and user engagement [50].

Validity evidence is equally strong. Content validity is rigorously confirmed through expert panel reviews and high Content Validity Index (CVI) scores, frequently above 0.90, ensuring items adequately represent the construct of nursing care quality [24,50]. Construct validity is supported through both exploratory and confirmatory factor analyses (EFA and CFA), which generally reveal factor structures explaining between 50% and 70% of total variance. CFA fit indices across studies commonly demonstrate good model fit, with Comparative Fit Index (CFI) values around 0.90 to 0.94 and Root Mean Square Error of Approximation (RMSEA) values ranging from 0.01 to 0.08 [50,54].

Factor structures of nursing care quality instruments vary from unidimensional scales to multi-dimensional models encompassing three to seven factors. These dimensions typically cover technical nursing competence, interpersonal relationships, patient comfort, psychological and spiritual care, empowerment, and organizational factors, reflecting the holistic nature of quality care [24,54]. Convergent validity evidence is mixed; some tools show moderate correlations (ranging approximately 0.3 to 0.6) with related measures like patient satisfaction or nurse competence, while others report weaker or inconsistent relationships, highlighting the complexity of capturing this multifaceted concept [53].

Face validity is commonly achieved through qualitative methods, including interviews with nurses and patients, alongside expert judgment, confirming the relevance and clarity of items in real-world practice [52]. Criterion and concurrent validity have been demonstrated in several instruments by correlating nursing care quality scores with external benchmarks or outcomes [50]. Some studies also incorporated qualitative research without formal psychometric testing, providing rich contextual insights into nursing care quality and complementing quantitative findings [54].

Cross-cultural adaptation and validation of these tools follow rigorous translation and back-translation procedures, ensuring linguistic and conceptual equivalence across diverse settings such as Iran, Cambodia, Mongolia, Portugal, and Turkey. This supports the global applicability and relevance of nursing care quality instruments.

Additional analyses using regression models identified burnout and work intensity as significant predictors negatively impacting nursing care quality. Models explaining variance in job outcomes, nurse fatigue, and perceived quality of care accounted for approximately 30% to 50% of the variability, indicating these work environment factors play a crucial role in shaping care quality.

### 3.5. Instrinsic Instrument Limitations

A self-report bias can be seen as an important limitation of Karen-Personnel/Karen Patient [15,19,23,33] and Quality Nursing Care Scale instruments, since they specifically rely on nurses’ self-assessment, which may introduce social desirability bias, overestimation of performance, or underreporting of missed care. Patient-reported measures are limited in some instruments, reducing triangulation of perspectives.

Furthermore, there are some instruments (Palliative Nursing Care Quality Scale and Good Nursing Care Scale) [27,34,35] that are very vulnerable to context, since they were developed in specific cultural, linguistic, or clinical contexts (e.g., oncology, ICU, hospital-based care). This can limit their generalizability across different healthcare settings or populations without rigorous adaptation and cross-cultural validation.

Instruments like SERVQUAL [14] or QONCS [24] have a static measurement limitation, since they often capture a single time-point, failing to reflect the dynamic and evolving nature of care quality. They may not account for fluctuations in staffing, patient acuity, or organizational changes that influence quality.

GNCS [27,34,35] has also an important ceiling and floor effect, where high-performing wards or highly experienced nurses may produce clustered scores at the top end (ceiling effect), reducing discriminatory power. Conversely, instruments may fail to detect poor-quality care in low-scoring environments (floor effect).

An important limitation would also be the length and complexity, particularly evident in the Karen-Personnel instrument [15,19,23,33] and QONCS [24]. Long multidimensional instruments may lead to respondent fatigue, incomplete responses, or reduced feasibility in busy clinical settings, impacting data quality.

Finally, MMSS and SERVQUAL [14] might suffer from a lack of integration with outcomes, since they measure perceived quality rather than linking scores directly to patient outcomes, limiting their predictive utility for evidence-based quality improvement.

## 4. Discussion

Nursing care quality (QNC) is characterized by a clear progression from early, largely unidimensional and outcome-focused measures toward comprehensive, multidimensional, and patient-centered frameworks. The enduring influence of Donabedian’s [6] Structure-Process-Outcome (SPO) model remains foundational across many instruments, providing a robust conceptual backbone for evaluating care quality in diverse settings, including hospitals and community care [26,30].

A key finding is the convergence between nurse and patient perspectives on the centrality of interpersonal dimensions such as empathy, communication, and respect as core components of quality nursing care. This is consistent with earlier studies [9,42] and aligns with person-centered care models that emphasize relational and emotional aspects as critical to patient satisfaction and safety [55]. Instruments such as the Karen-patient and Karen-personnel scales [15], the Good Nursing Care Scale (GNCS) [35,54], and the QONCS [24,49] embody this multidimensionality by incorporating affective care, spiritual support, and patient-staff relationship domains.

Cross-cultural validations of tools such as the GNCS, QONCS, and CNCQS demonstrate their broad international applicability, while also underscoring the importance of rigorous cultural adaptation. Studies from Mongolia [34,37], Cambodia [31], Iran [44], and Turkey [40,50] illustrate a growing commitment to improving nursing care quality through context-sensitive assessment instruments.

Organizational and structural factors emerge as critical determinants of perceived care quality. Consistent associations between supportive leadership, adequate staffing levels, and positive work environments with higher quality ratings highlight the organizational context’s influence [22,29,53]. Empowering work environments have been shown to improve professional behaviors, enhance job satisfaction, and mitigate burnout and missed care, as evidenced in recent studies from Thailand and China [43,47].

Psychometrically, the reviewed instruments exhibit strong reliability—internal consistency measures such as Cronbach’s alpha frequently exceed 0.90, with test–retest reliability and inter-rater reliability also demonstrating excellent stability and agreement. Validity is well supported through expert-reviewed content validity (CVI > 0.90), construct validity confirmed via exploratory and confirmatory factor analyses explaining 50–70% of variance, and good model fit indices (CFI ~0.90–0.94, RMSEA ~0.01–0.08). Despite these strengths, convergent validity results are variable, reflecting the complexity of nursing care quality as a construct. Qualitative studies, though less common, provide complementary insights into contextual and relational aspects beyond the quantitative psychometric scope.

Nonetheless, several gaps remain. Longitudinal research assessing the sustained impact of quality improvement interventions is limited. Integration of patient-reported outcome measures (PROMs) remains insufficient, despite their potential to deepen the understanding of patient-centered quality. Additionally, while many tools demonstrate strong psychometric properties, few studies offer comprehensive guidance on implementation or culturally sensitive adaptation processes.

The findings carry significant practical implications for nursing practice, health systems, and the visibility of the profession in clinical contexts. As the assessment of nursing care quality becomes increasingly nuanced, healthcare organizations must prioritize the adoption of instruments that are both psychometrically robust and sensitive to specific cultural and clinical contexts. Widely validated tools such as the GNCS, QONCS, and Karen scales offer not only reliable measurement but also an opportunity to enhance the visibility of nursing contributions to patient outcomes when adapted with stakeholder participation. By systematically documenting the relational, technical, and organizational dimensions of care, these instruments make the often-invisible work of nurses more explicit, strengthening recognition of nursing’s role within multidisciplinary teams.

A central implication for clinical practice is the prioritization of person-centered care in both daily work and quality assessment. The recurrent emphasis on empathy, communication, and respect by both nurses and patients positions relational competencies as core indicators of high-quality care. This underscores the need for sustained investment in training, reflective practice, and continuing professional development that reinforce the caring dimension of nursing, which is frequently undervalued compared to technical proficiency.

At the organizational level, the findings highlight the decisive impact of supportive leadership, adequate staffing, and a healthy work environment on both care quality and professional sustainability. Investing in these structures not only improves patient safety and satisfaction but also enhances nurse well-being, motivation, and retention. In turn, this reinforces the profession’s visibility as a driver of patient-centered, safe, and equitable healthcare delivery.

Finally, the integration of validated instruments into routine evaluation processes provides a means for nurses to systematically demonstrate the value of their practice. By generating measurable evidence of the quality and impact of nursing care, these tools empower the profession to contribute more strongly to quality improvement initiatives, policy design, and public discourse, thereby elevating nursing’s profile as an indispensable component of healthcare systems worldwide.

Finally, incorporating PROMs and qualitative feedback into routine quality monitoring can enrich traditional metrics by capturing patient experiences, offering a more holistic and human-centered evaluation of nursing care. However, a major limitation revealed in this review is that only a few instruments systematically integrate PROMs into their design. Most tools continue to emphasize structural or process-related indicators (e.g., staffing ratios, adherence to protocols, technical competence), with limited attention to outcomes directly reported by patients, such as symptom relief, emotional support, dignity, and overall satisfaction. For instance, while instruments like the QONCS and PNCQS begin to address patient perspectives, these remain exceptions rather than the norm. This gap constrains the ability of current measurement systems to reflect what matters most to patients in their lived experiences of care. Future instrument development and refinement should therefore explicitly embed PROMs as central components, ensuring that the quality of nursing care is evaluated not only through organizational or professional lenses but also through the voices of those receiving care.

Future research should embrace mixed-methods and implementation science to unravel the complex interrelations among nurse well-being, patient outcomes, and organizational culture. There is also an urgent need to develop and validate tools that are sensitive to emerging challenges, such as digital health integration, equity in care delivery, and the critical role of interprofessional collaboration in improving the quality of nursing care.

One of the main limitations of the chosen method and eligibility criteria is the inclusion of both quantitative and qualitative studies introduced methodological heterogeneity that may affect the consistency of data synthesis. While this allowed for a richer conceptual mapping, it also made comparison across studies more complex, particularly in relation to psychometric properties and instrument performance. Furthermore, some included studies lacked full psychometric validation or reported limited statistical details, which may affect the reliability of conclusions regarding the robustness of certain instruments. Additionally, some qualitative studies were included to inform conceptual development but did not result in concrete tools, which may reduce comparability with instrument-based assessments.

## 5. Conclusions

This scoping review highlights the dynamic, multidimensional nature of nursing care quality (QNC) assessment, reflecting a global commitment to advancing care delivery through evidence-based, culturally sensitive, and contextually relevant instruments. The sustained prominence of Donabedian’s Structure-Process-Outcome model attests to its foundational role, while the increasing focus on interpersonal and relational dimensions—such as empathy, communication, and respect—signals a progressive shift toward holistic, person-centered frameworks.

The review further identifies a growing array of psychometrically sound instruments that have been successfully adapted and validated across diverse cultural and clinical settings, including in low- and middle-income countries. Nevertheless, critical gaps persist, particularly in the longitudinal evaluation of quality improvement initiatives, the systematic incorporation of patient-reported outcome measures (PROMs), and the nuanced exploration of nurse well-being as a key determinant of care quality.

Bridging these gaps will require interdisciplinary collaboration, innovative mixed-methods research, and sustained investment in nursing science and practice development. These efforts are vital to fully grasp the complexity of nursing care quality and to continuous improvements in care experiences and outcomes globally.

## Figures and Tables

**Figure 1 nursrep-15-00342-f001:**
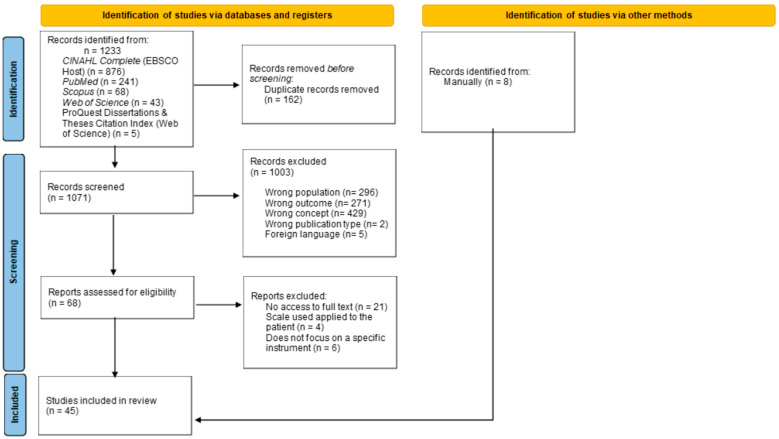
PRISMA-Scr Flowchart.

**Table 1 nursrep-15-00342-t001:** Synthesis of the findings (n = 45).

Author(s)/Year	Country	Instrument(s)	Definition of Quality	Sample and Context	Key Findings
Redfern & Norman/1995 [11]	UK	Monitor, Senior Monitor, Qualpacs	Aligns with the nursing process and incorporates autonomy, documentation, and PCC.	11 wards (medical-surgical, elderly care).	Senior Monitor and Qualpacs are suitable for assessing quality in elderly care. Monitor DG3 is useful for high-dependency patients, but other subscales lack validity. Implementation challenges were noted across tools.
Redfern & Norman/1996 [12]	UK	Monitor, Senior Monitor, Qualpacs	Four domains: care planning, physical care, non-physical care, and evaluation. Influenced by patient dependency, omitted care activities, and congrunce between nurse/patient perceptions.	11 wards (medical-surgical, elderly care). 123 patients and 80 nurses.	Senior Monitor demonstrated better validity than Monitor. Monitor should be used as four separate schedules (DG1–DG4), with DG3 showing strongest performance. Approximately 10% of items lacked endorsement and may be unnecessary.
Mrayyan/2006 [13]	Jordan	Mueller/McCloskey Satisfaction Scale (MMSS); Eriksen’s Satisfaction with Nursing Care Questionnaire; Quality of Nursing Care Questionnaire—Head Nurse	Care provided according to hospital standards and job requirements.	200 nurses, 510 patients, and 26 head nurses in one educational hospital	Nurses reported neutral satisfaction levels; patients were moderately satisfied. Head nurses rated care quality as generally high. Positive correlations were observed between nurse job satisfaction and patient satisfaction.
Lee & Yom/2007 [14]	South Korea	SERVQUAL (adapted and translated into Korean)	Defined as the gap between expectations and performance across five dimensions: tangibility, reliability, responsiveness, assurance, and empathy.	272 patients and 282 nurses from six hospitals in Korea	Nurses had higher expectations and performance ratings, while patients reported higher satisfaction. A significant expectation-performance gap indicated poor perceived quality. Satisfaction strongly correlated with intent to revisit. Results suggest a need for managerial strategies to align staff and patient perspectives.
Andersson & Lind/2008 [15]	Sweden	Karen-patient and Karen-personnel instruments (part of the KISAAL system)	Based on Donabedian’s model (Structure-Process-Outcome), quality includes staff characteristics, affective care, patient-staff relationships, and patient outcomes (e.g., satisfaction, health improvement).	Conduced in a Swedish university hospital with 64 patients and 42 staff members from a medical-surgical ward.	The reduced instruments retained conceptual integrity and demonstrated good reliability. They allow comparison between patient and staff perspectives on structure and process quality, though not outcome quality.
Zhao et al./2009 [16]	China	Perception of Quality Nursing Care Scale (PQNCS)	Six categories: staff characteristics, care-related activities, preconditions for care, physical environment, progress of nursing process, and cooperation with relatives.	Conducted in 18 non-ICUs at a tertiary hospital in Harbin, China, with 221 nurses and 383 patients.	Both groups rated care quality as high, but significant differences were found in perceptions of staff characteristics, care-related activities, and nursing process. Cultural factors and expectations influenced patient responses
Burhans & Alligood/2010 [9]	US	No formal instrument; qualitative study using hermeneutic phenomenology	Quality nursing care is defined as meeting human needs through caring, empathetic, respectful interactions, grounded in responsibility, intentionality, and advocacy.	12 female registered nurses from acute care hospitals in southeastern USA.	The essence of quality nursing care lies in six themes: responsibility, caring, intentionality, empathy, respect, and advocacy. These reflect the art of nursing and are consistently recognized by nurses in their own and others’ practice. Findings suggest that incorporating these themes into education and management could improve care quality.
Maes et al./2010 [17]	France	IGEQSI (Instrument Global d’Évaluation de la Qualité des Soins Infirmiers)	Not explicitly defined; focus on affective commitment and satisfaction as outcomes of quality initiatives.	30 nurses and nursing assistants in a French clinic.	Job satisfaction is linked to professional experience and value alignment. Affective commitment is fostered by autonomy, recognition, and team cohesion. Implementation of quality tools can enhance engagement and satisfaction.
Donmez & Ozbaır/2010 [18]	Turkey	Good Perioperative Nursing Care Scale (GPNCS)—Turkish version	Includes physical care, giving information, support, respect, personnel characteristics, environment, and nursing process.	346 patients and 159 nurses from 11 hospitals in Turkey.	The Turkish GPNCS is a valid and reliable tool for assessing perioperative nursing care. It supports both patient and nurse perspectives and is suitable for clinical quality improvement.
Lindgren et al./2011 [19]	Sweden	Karen-Patient and Karen-Personnel Instruments	Based on Donabedian’s Structure–Process–Outcome (SPO) model and qualitative interviews with patients and staff.	Conducted in medical and surgical wards in a Swedish hospital. 95 patients and 120 staff (nurses and nursing aides) participated.	The Karen instruments demonstrate good construct validity and internal consistency. Suitable for clinical use to assess and compare patient and staff perceptions of nursing care quality.
Cline et al./2011 [20]	USA	None (qualitative content analysis of narrative responses)	Quality care is defined through three themes: RN presence, developing relationships, and facilitating the flow of knowledge and information.	171 narrative responses from early-career hospital-based RNs in the U.S., as part of a longitudinal survey.	High-quality nursing care was described as relational and process-oriented, focusing on presence, trust-building, and evidence-based communication. The study underscores the need for quality indicators that go beyond outcomes to reflect core nursing processes.
Wilson et al./2012 [21]	Australia	No single instrument; 57 indicators developed and refined using a modified Delphi process.	Framed using Donabedian’s model: Structure: Staffing levels, skill mix; Process: Pain management, assessments; Outcome: Pressure ulcers, infections, unplanned extubation	Modified Delphi study with 52 pediatric nursing experts (clinicians, educators, managers) across Australia, using three rounds of surveys.	42 indicators validated for pediatric care; traditional adult indicators (e.g., mortality, DVT) deemed unsuitable. Data collection from case notes was often difficult. Ongoing research is recommended to evaluate indicator utility and frequency in real-world settings.
Bogaert et al./2013 [22]	Multicentric	Revised Nursing Work Index (NWI-R) Maslach Burnout Inventory (MBI) Intensity of Labour Scale Decision Latitude and Social Capital Scales	Nurses’ self-assessed quality at the unit, shift, and hospital levels, influenced by collaboration, management support, and workload.	Cross-sectional survey of 1201 nurses across 9 hospitals in Belgium (general, university, and hospital group settings).	Strongest predictor of job satisfaction and quality perception was unit-level nurse management. Workload and burnout mediated negative effects, while decision latitude and social capital buffered against burnout. Empowered, collaborative environments enhance care quality.
Andersson & Lindgren/2013 [23]	Sweden	Karen-patient and Karen-personnel instruments	Grounded in Donabedian’s model and patient-centered care. Quality assessed across subscales: satisfaction, influence, staff competence, caring/uncaring, integrity, and organization.	Swedish regional hospital; 95 patients and 120 nursing staff (registered and assistant nurses).	Both groups rated care quality positively overall. Patients scored staff competence highly but expressed lower satisfaction with organizational factors like continuity of care. The Karen tools effectively highlight perceptual gaps and areas for improvement.
Charalambous & Adamakidou/2014 [24]	Cyprus and Greece	Quality of Oncology Nursing Care Scale (QONCS)	Holistic concept including five dimensions: support and confirmation, spiritual caring, sense of belonging, being valued, and being respected.	Multicenter study in Cyprus and Greece with 418 hospitalized cancer patients. Used a mixed-methods approach: literature review, expert input, pilot study, and large-scale validation.	QONCS is a valid and reliable instrument that captures a holistic, patient-centered view of oncology nursing care. It uniquely integrates spiritual and emotional aspects, addressing key gaps in existing tools.
Rossaneis et al./2014 [25]	Brasil	Electronic questionnaire developed by the authors	Not explicitly defined; focused on the presence and use of quality indicators in practice.	Nine teaching hospitals in Paraná, Brazil; participants were nurse managers.	While nursing quality indicators are commonly used, they lack standardization and are not benchmarked across institutions. The study highlights the need for unified strategies to evaluate, compare, and improve nursing care quality in varied contexts.
Voyce et al./2015 [26]	Portugal	Adapted questionnaire based on Donabedian’s model.	Structured around Donabedian’s three components: Structure: Resources, facilities, organization; Process: Nurse-patient interactions; Outcome: Results like satisfaction and health status.	Emergency department (obstetrics/gynecology) in Algarve Hospital Centre, Portimão, Portugal. Sample: 23 nurses.	Donabedian’s model was supported as a valid structure for assessing nursing care quality, despite moderate internal consistency. Process and outcome domains showed perceptual overlap. The study highlights the need for robust statistical techniques—especially for small samples—to explore multidimensional quality constructs.
Koy et al./2016 [27]	Thailand	Good Nursing Care Scale (GNCS), Karen-patient and Karen-personnel, Patient Perception of Hospital Experience with Nursing (PPHEN), Nurses’ Assessment of Quality Scale (NAQS-ACV)	Varies by perspective—nurses emphasize competence and empathy; patients prioritize responsiveness and communication. The review highlights NCQ as a multidimensional concept.	Systematic review of 18 studies conducted across the USA, Europe, Asia, and Canada.	No single universal instrument exists. Perceptions of quality differ between patients and nurses, emphasizing the importance of selecting tools that suit the specific context and purpose. Greater consensus and standardization in NCQ measurement are needed.
Martins et al./2016 [28]	Portugal	Escala de Perceção das Atividades de Enfermagem que Contribuem para a Qualidade dos Cuidados (EPAECQC)	Aligned with standards from the Portuguese Order of Nurses, covering seven domains: client satisfaction, health promotion, complication prevention, well-being and self-care, functional readaptation, organization of care, and professional responsibility.	Hospital in northern Portugal; 775 nurses participated (May–July 2014).	The EPAECQC is a robust, valid, and reliable instrument. It reflects national care quality standards and is applicable for both research and practical improvements in nursing care.
Laschinger et al./2016 [29]	Canada	Conditions for Work Effectiveness Questionnaire (CWEQ) Nursing Work Index-Revised (NWI-R) Professional Practice Behaviours Scale (developed for the study) Single-item measure for perceived care quality	Based on nurses’ perceptions of their ability to deliver care aligned with professional standards. Influencing factors include empowerment, autonomy, practice control, and collaboration.	National survey of 393 new graduate nurses in Canada (within their first 3 years of practice).	Work environments that empower and support professional practice improve nurses’ behaviors, perceived care quality, job satisfaction, and retention. Organizational strategies fostering empowerment and collaboration are crucial for sustaining quality nursing care.
Alcântara-Garzin & Melleiro/2017 [30]	Brazil	Custom-built Likert-scale tool; Based on Donabedian’s model: Structure: Resources, infrastructure; Process: Nursing activities, interpersonal relationships; Outcome: Service characteristics, patient satisfaction, safety	Quality is seen as the interaction of structure, process, and outcomes, with an emphasis on the ratio between service effectiveness and user expectations. The goal is to ensure a safe environment that minimizes risks to users.	Private diagnostic medicine institution, São Paulo, Brazil Participants: 203 nursing professionals	The validated instrument is reliable and useful for guiding managerial and clinical actions to improve nursing care quality in diagnostic medicine.
Koy et al./2017 [31]	Cambodia	Cambodian Nursing Care Quality Scale (CNCQS); Measures the degree to which nursing activities meet professional standards and patient needs, as perceived by nurses.	The degree to which nursing activities meet professional standards and patient needs, as perceived by nurses.	225 registered nurses from 12 hospitals across Cambodia.	CNCQS is a valid and reliable tool for assessing nursing care quality in Cambodia; Further testing is recommended across diverse settings; The tool supports quality improvement and professional development in nursing.
Martins et al./2017 [32]	Portugal	Escala da Perceção das Atividades de Enfermagem que Contribuem para a Qualidade dos Cuidados	Nursing care quality based on Ordem dos Enfermeiros’ standards emphasizing patient satisfaction, health promotion, prevention, well-being, functional readaptation, organization, responsibility, and rigor.	36 public hospitals across mainland Portugal; 3451 nurses.	Nurses commonly implement responsibility and prevention activities but less frequently health promotion, self-care, and functional readaptation, indicating a need for practice redesign and additional training.
Alsyouf et al./2018 [33]	Jordan	Karen-personnel instrument	Quality in psychiatric nursing care includes psychosocial relationships, commitment, job satisfaction, openness/proximity, competence development, and safety/insecurity; quality indicators differ from other health services and require continuous observation and measurement.	Psychiatric inpatient units in Jordan.	64% of nurses rated psychiatric nursing care as satisfactory; Nurses generally perceive the quality of care more positively than patients. Highlights differing perceptions between nurses and patients and the unique challenges in measuring psychiatric nursing care quality.
Gaalan et al./2019 [34]	Mongolia	Good Nursing Care Scale (GNCS)	Quality defined as excellence in addressing patients’ physical, psychological, emotional, social, and spiritual needs (Leino-Kilpi model).	346 registered nurses from seven tertiary public hospitals in Ulaanbaatar and four other regions.	Nurses with higher competence and positive practice environments deliver better nursing care quality.
Stolt et al./2019 [35]	Finland	Good Nursing Care Scale (GNCS)	Quality is based on Donabedian’s model (structure, process, outcome) and action theory, including safety, effectiveness, patient-centeredness, and nurse competence.	One university hospital with 480 surgical patients and 167 nurses.	GNCS is a valid and reliable instrument for assessing nursing care quality from both patient and nurse perspectives, though some items may need revision for improved fit.
Egea et al./2020 [36]	Spain	Palliative Nursing Care Quality Scale (PNCQS)	Quality is defined as holistic, patient- and family-centered palliative care, including symptom control, therapeutic relationships, spiritual support, and continuity of care. It emphasizes personal values such as growth, dedication, and the meaningfulness of nursing.	Stage 1 involved qualitative interviews with 10 key informants in Spain; Stage 2 included 100 nurses from Madrid; Stage 3 surveyed 176 nurses from various palliative care centers across the country.	The PNCQS is a valid and reliable instrument for evaluating the quality of palliative nursing care. It supports professional autonomy and provides a foundation for continuous improvement in care practices.
Tsogbadrakh et al./2021 [37]	Mongolia	Quality Nursing Care Scale—Mongolia (QNCS-M)	Quality nursing care is defined as the provision of physical, psychological, emotional, social, and spiritual care, shaped by cultural norms, patient needs, and the healthcare environment.	Phase I involved qualitative development; Phase II included validation with 440 nurses from 9 public hospitals in Ulaanbaatar.	The QNCS-M is a culturally relevant, valid, and reliable instrument for evaluating nursing care quality in Mongolia and can be adapted for use in comparable healthcare contexts.
Moen et al./2021 [38]	Norway	QPC-COPS (Quality in Psychiatric Care—Community Outpatient Psychiatric Staff), FINC-NA (Families’ Importance in Nursing Care—Nurses’ Attitudes), SOC-13 (Sense of Coherence Scale, 13-item version).	Quality of care is assessed across eight dimensions: Encounter, Participation (Empowerment and Information), Discharge, Support, Environment, Next of Kin, Accessibility. It is influenced by attitudes toward family involvement and the professional’s sense of coherence (SOC).	Cross-sectional quantitative study with 56 community mental health professionals (primarily nurses) from 17 municipalities in Norway, all with at least one year of experience.	Overall quality of care was rated high, especially in the “Encounter” dimension. Family involvement was not seen as a burden, but their participation as conversational partners was limited. Professionals with higher SOC scores reported higher perceived care quality and fewer negative views of family involvement. Interestingly, longer work experience correlated with lower ratings of care quality, and nurses had more positive attitudes toward family involvement compared to other professionals.
Liu et al./2021 [39]	China	Quality Nursing Care Scale (QNCS)	Quality nursing care was conceptualized through six key dimensions: team characteristics, task-oriented activities, human-oriented activities, physical environment, patient outcomes, and care preconditions, reflecting both technical and relational aspects of care.	302 nurses participated through random sampling.	The QNCS is a valid and reliable instrument for evaluating nursing care quality from the perspective of nurses in the Chinese healthcare context.
Karaca et al./2022 [40]	Turkey	Quality Nursing Care Scale (QNCS)—Turkish version	Quality is defined from the perspective of nurses and includes dimensions such as the physical environment, staff characteristics, nursing tasks, and patient outcomes.	225 nurses from a training and research hospital in Turkey.	The Turkish adaptation of the QNCS is a valid and reliable tool for assessing nursing care quality from nurses’ perspectives and can effectively support clinical practice improvement in Turkish healthcare settings.
Malakeh et al./2022 [41]	Jordan	Karen-personnel instrument	The instrument assesses quality of nursing care through six subscales: psychosocial relationships, commitment, job satisfaction, openness/proximity, competence development, and safety/insecurity, reflecting nurses’ perceptions of the relational and organizational aspects of care.	Intensive Care Units (ICUs)	Psychological empowerment at work is significantly associated with improved nursing care quality. Enhancing empowerment among ICU nurses is necessary to support higher standards of care.
Stavropoulou et al./2022 [42]	Greece	No formal instrument; qualitative descriptive study using semi-structured interviews	Quality nursing care is defined as holistic care that meets patient needs, achieves optimal outcomes, and is grounded in communication, teamwork, leadership, and personal commitment.	10 female clinical nurses from a public hospital in Athens.	Four core themes were identified: quality care is holistic, good care depends on interpersonal relationships, leadership plays a central role, and personal responsibility drives care quality. Nurses highlighted empathy, teamwork, and leadership support as critical components, suggesting a need for stronger organizational and educational strategies to promote holistic, high-quality nursing care.
Nantsupawat/2023 [43]	Thailand	Maslach Burnout Inventory—Human Services Survey (MBI-HSS), MISSCARE Survey, and two single-item Likert-scale measures for perceived quality of care	Quality of care was self-reported by nurses using Likert-scale questions at both unit and shift levels; burnout was defined as emotional exhaustion (EE score ≥ 27), and missed care as any care activity reported as missed with any frequency.	Cross-sectional survey of 394 nurses from 12 general hospitals in Thailand, conducted between August and October 2022; all participants had at least one year of nursing experience during the COVID-19 pandemic.	Burnout was strongly associated with increased missed care and lower perceived quality of care. Specifically, burnout raised the odds of missed care by 1.61 times, poor care during the last shift by 3.37 times, and poor overall unit care by 2.62 times. Emotional exhaustion remains a major concern post-pandemic, underscoring the need for organizational support strategies such as improved staffing and access to mental health resources.
Tehranineshat et al./2023 [44]	Iran	Nursing Care Quality Scale—Persian (CNCQS-Persian)	Rooted in Donabedian’s model (Structure–Process–Outcome), quality is defined through six domains: patient outcomes, ethics-oriented activities, nurses’ characteristics, task requirements, nursing process, and physical environment.	310 nurses from four teaching hospitals in Iran; data collected between May 2021 and March 2022.	The CNCQS-Persian is a psychometrically sound instrument for assessing nursing care quality in Iran. Its culturally adapted structure allows for a comprehensive evaluation of quality in line with local clinical practices and healthcare expectations.
Hong et al./2023 [45]	South Korea	Korean Patient Classification System—General Ward (KPCS-GW), Work Sampling Tool, and an Ad Hoc survey for perceived staffing adequacy, fatigue, and care quality	Quality of care was self-rated by nurses using a 4-point Likert scale, considered in relation to perceived staffing adequacy and fatigue. Staffing adequacy was assessed on a scale from −10 to +10, fatigue on a 0–10 scale. Nurse staffing was measured Via three approaches: nurse-to-patient ratio, acuity-adjusted work intensity, and demanded nursing hours per nurse.	Cross-sectional study conducted in a general hospital across 6 wards, including 90 nurses and 5536 patients; data collected daily over a 4-week period in 2022.	The study found that nurse-to-patient ratios are insufficient for assessing staffing adequacy. Acuity-adjusted metrics such as work intensity and nursing hours per patient day are better predictors of nurse fatigue and perceived care quality. Integrating these measures into staffing models can enhance care quality and reduce nurse burnout.
Román/2023 [46]	Cuba	Instrument to measure the perceived quality of nursing services in the hospital context	Quality is a complex concept beyond technical and mechanical aspects, involving human care, empathy, and integration of values and scientific knowledge. It is based on Donabedian’s model with three dimensions: Technical (adherence to standards, technical ability), Interpersonal (nurse-patient relationship, communication, ethics), and Comfort (physical and emotional environment).	Hospital setting involving 9 experts, 15 judges, 30 hospitalized patients, and 10 nursing professionals.	The instrument demonstrated high content validity and reliability, allowing a holistic evaluation of perceived nursing care quality from both patient and professional viewpoints. It is useful to identify gaps in care and to improve nursing quality in hospitals.
Xue et al./2023 [47]	China	Good Nursing Care Scale (GNCS), Practice Environment Scale (PES-NWI), Utrecht Work Engagement Scale (3-UWES), Psychological Empowerment Scale (PES), High-Performance Work Systems Scale (HPWSS), Perceived Organizational Support (8-SPOS)	Quality of nursing care is defined as the degree of excellence that meets patients’ spiritual, mental, social, physical, and environmental needs.	784 nurses from three university-affiliated hospitals in China.	Practice environment, psychological empowerment, and work engagement were found to have significant positive effects on nursing care quality. The proposed causal model is promising but requires further testing and refinement to confirm its applicability.
Rivaz & Tehranineshat/2023 [48]	Iran	Professional Collaboration Subscale (from the Professional Practice Environment Nursing Instrument), Nursing Care Quality Scale (NCQS)	Quality of nursing care is defined as the ethical, safe, and effective delivery of care, with professional collaboration—particularly ethics-oriented activities—being a key predictor.	Cross-sectional study with 330 ICU nurses in Shiraz, Iran.	Professional collaboration was significantly associated with higher quality nursing care, with ethics-oriented collaboration having the strongest impact. The study suggests that educational programs focusing on professional ethics could enhance care quality in ICU settings.
Marcomini et al./2024 [49]	Italy	Quality of Oncology Nursing Care Scale (QONCS)—Italian version	Quality is defined as patients’ subjective perception of nursing care, including professional support, spiritual respect, sense of belonging, feeling valued, and being respected.	Cross-sectional study involving 219 oncology patients from three hospitals in Northern and Central Italy.	The Italian QONCS is a valid and reliable tool for comprehensively assessing nursing care quality in oncology settings. Patient factors such as age, marital status, and employment status were found to influence perceived quality of care.
Mollaoğlu et al./2024 [50]	Turkey	Quality Nursing Care Scale—Turkish version (QNCS-T)	Quality is understood as holistic care, encompassing psychological, spiritual, social, and professional dimensions.	347 nurses from a university hospital in Turkey participated in the study.	The QNCS-T is a valid and reliable instrument for assessing nursing care quality in Turkey, supporting a holistic evaluation approach and serving as a useful tool for guiding nurse training and quality improvement initiatives.
Toptaş et al./2024 [51]	Turkey	Palliative Nursing Care Quality Scale—Turkish version (PNCQS-TR)	Holistic palliative nursing care encompassing symptom management, communication, ethical responsibility, family involvement, and continuity of care.	Methodological study with 210 palliative care nurses from various units including oncology, ICU, and geriatrics in Turkey; data collected online from September to December 2021.	The PNCQS-TR is a valid and reliable instrument for assessing palliative nursing care quality in Turkey. It facilitates self-assessment among nurses and can be used to support quality improvement efforts in clinical practice.
Veitamana et al./2024 [52]	Fiji Islands	Quality of Care Scale (QOCS) developed by Aiken and Patrician	Nursing care quality as a critical factor influencing healthcare service success, measured by nurses’ perceptions of care quality on a 4-point scale from poor to excellent, affected by nurse shortages, workload, and work conditions.	Cross-sectional descriptive-predictive study involving 744 registered nurses in three tertiary hospitals in Fiji.	72.58% of nurses rated overall care quality as good or excellent. Relational coordination and job satisfaction were significant predictors positively influencing nurses’ perceptions of care quality, highlighting the importance of interpersonal and organizational factors in perceived nursing care quality.
Liu et al./2024 [53]	China	Palliative Nursing Care Quality Scale (PNCQS)—Chinese version	Quality is defined as overall palliative nursing care quality encompassing symptom management, communication, ethical responsibility, family involvement, and continuity of care.	Mainland China, palliative nursing care context.	The Chinese version of PNCQS is valid and reliable for assessing palliative nursing care quality in mainland China.
Mattila et al./2025 [54]	Finland	Good Nursing Care Scale (GNCS)	Patient-centered care encompassing nurse characteristics, care activities, environment, care process, empowerment, and family collaboration.	Systematic review across multiple countries and care settings (surgical, pediatric, ICU, etc.).	The GNCS is a valid and reliable tool for assessing patient-centered nursing care internationally and is suitable for long-term quality monitoring in diverse healthcare settings.

PCC: Patient-Centered Care.

## Data Availability

All detailed data is available upon author’s request.

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
