# Peer review of "Instruments for Assessing Nursing Care Quality: A Scoping Review"

_nursrep, 2025, doi:10.3390/nursrep15090342_

Round 1

Reviewer 1 Report

Comments and Suggestions for Authors

Thank you for the opportunity to review the paper. This is an important research to fill the gap in nursing research. My only suggestion is to add some limitations to the included articles in the table, if applicable. Spell out Open Science Framework (OSF). 

The paper synthesized the available research instruments that can be used to measure nursing care quality. This is an important topic in healthcare. The paper is well-written and follows the methodological process of reviewing articles using PRISMA with appropriate search strategy- keywords/MESH words. The tabulated data extraction sheet provides adequate information about the included articles, although authors can add a column to indicate the limitation of use for each tool. The following are some minor comments:
  1. Spell out OSF.
  2. Cite the reference for PRISMA-ScR

Author Response

We have attached the responses in the PDF file.

Reviewer 2 Report

Comments and Suggestions for Authors

The manuscript is generally well-written and adheres to a high standard of academic English. However, there are a few minor issues that could be addressed to further improve clarity and readability:

  1. Grammar and Syntax:

    • Some sentences are overly complex or lengthy, which could be simplified for better readability. For example, the sentence beginning with "The sustained prominence of Donabedian’s Structure-Process-Outcome model attests to its foundational role..." could be broken into shorter sentences.

    • Ensure consistent verb tense usage, especially in the Results and Discussion sections.

  2. Word Choice and Precision:

    • The term "booth" in the phrase "encompassing booth hospital units" (Page 3) appears to be a typo and should be corrected to "both."

    • The phrase "illustrating the diversity of nursing roles and cultural contexts worldwide" (Page 20) could be reworded for smoother flow, such as "highlighting the diversity of nursing roles across cultural contexts worldwide."

  3. Clarity and Conciseness:

    • Some sentences could be more concise. For example, "The review further identifies a growing array of psychometrically sound instruments..." (Page 23) could be simplified to "The review also identifies psychometrically sound instruments..."

    • Avoid redundancy, such as "critical gaps persist, particularly in the longitudinal evaluation of quality improvement initiatives" (Page 23), where "particularly" could be omitted without losing meaning.

  4. Formatting and Consistency:

    • Ensure consistent formatting of references and in-text citations. For example, some references use "et al." in italics, while others do not.

    • Check for consistent use of hyphens in compound terms (e.g., "patient-centered" vs. "patient centered").

  5. Minor Typographical Errors:

    • "The Karen Institute demonstrates good construct" (Page 7) appears to be an incomplete sentence or heading.

    • "Coldibility of nursing care" (Page 18) seems to be a typo and should likely be "Quality of nursing care."

Comments on the Quality of English Language

Grammar and Syntax

  • Page 3: "encompassing booth hospital units" → Correct to "both hospital units."

  • Page 7: "The Karen Institute demonstrates good construct" → This appears incomplete; revise for clarity (e.g., "The Karen instruments demonstrate good construct validity").

  • Page 18: "Coldibility of nursing care" → Likely a typo; correct to "Quality of nursing care."

Clarity and Conciseness

  • Page 20: "illustrating the diversity of nursing roles and cultural contexts worldwide" → Consider rewording for smoother flow (e.g., "highlighting the diversity of nursing roles across global contexts").

  • Page 23: "The review further identifies a growing array of psychometrically sound instruments..." → Could be simplified to "The review also identifies psychometrically sound instruments..."

Word Choice and Precision

  • Page 23: "critical gaps persist, particularly in the longitudinal evaluation..." → The word "particularly" is redundant here and could be omitted.

  • Page 20: "Quality is also framed as the gap between patient expectations and the actual delivery of care" → Consider "Quality is also defined by the gap between patient expectations and care delivery."

Consistency

  • Ensure uniform hyphenation (e.g., "patient-centered" vs. "patient centered").

  • Check for consistent use of "et al." (italicized vs. plain text in references).

Minor Typographical Errors

  • Page 6 (Table 1): "Pre-conditions for care" → Should likely be "Preconditions for care."

  • Page 22: "Cross-cultural adaptation and validation of these tools follow rigorous translation and back-translation procedures" → Consider "These tools underwent rigorous cross-cultural adaptation, including translation and back-translation."

Author Response

(The authors gave the same response as above.)

Reviewer 3 Report

Comments and Suggestions for Authors

Thank you for the opportunity to review the manuscript "Instruments for Assessing the Quality of Nursing Care: A Scoping Review."

Strengths

The methodological rigor of this review is evident. It follows the methodology of the Joanna Briggs Institute (JBI) and is reported in accordance with PRISMA-ScR, ensuring transparency and robustness. Registering the protocol in OSF further strengthens reproducibility.
The search strategy is exhaustive and includes recognized databases (CINAHL, PubMed, Scopus, Web of Science, ProQuest) and reports the existence of gray literature.

It demonstrates intercultural coverage and adaptation of instruments, providing a global perspective on the quality of nursing care. There is analysis of technical and interpersonal dimensions of attention, including empathy, communication and respect, psychometric properties (reliability, validity). The review highlights organizational factors (leadership, personal burnout) as critical determinants of perceived quality of care and the review identifies applicable instruments (GNCS, QONCS, Karen scales).

Weaknesses and areas for improvement

The research questions that guide the results could be addressed more explicitly; this has to be a PICO question that encompasses the results.  While the manuscript clearly responds to the context of the studies, the psychometric properties, and the instruments available, the discussion of the domains seems weak. Since nursing conceptualizes domains based on the nature of care, I recommend reviewing and specifying this aspect. If the authors prefer not to reduce the guiding question, I would at least recommend providing a comparative synthesis of the recurring dimensions across all instruments (empathy, communication, leadership, environment), which would strengthen the conceptual objectivity of the findings.

The theoretical basis of the review could be reinforced by incorporating conceptual frameworks complementary to Donabedian's model (e.g., Watson's Theory of Human Care, Leininger's Theory of Transcultural Nursing, or whatever they consider pertinent according to what was found in the study). This would allow interpretation of the findings beyond psychometric analysis of instruments.

It would be valuable to address potential cultural biases and limitations in cross-cultural adaptations of the instruments, highlighting how this review not only describes the tools available, but also offers a practical framework to guide the selection of instruments in various clinical and research contexts.

It is important to carry out a section of practical implications regarding the quality of nursing care and the visibility of this profession in clinical contexts.

Although the manuscript highlights the importance of PROMs, few instruments incorporate them systematically. This limitation should be discussed more explicitly.

Finally, the presentation could benefit from greater visual synthesis in the tables to enhance readability.  But I understand that it is a challenge because of the number of instruments, I leave it to your consideration.

Please check very well because there are inconsistencies in the reporting of the included studies. The manuscript states that 59 studies were included, the PRISMA diagram reports 58, but only 55 are listed in the reference list. These discrepancies must be corrected accurately.

Author Response

(The authors gave the same response as above.)

Reviewer 4 Report

Comments and Suggestions for Authors

The review is highly relevant and timely related to instruments that measure nursing care quality. The overall structure and science of the manuscript are good. However, a number of issues need to be fixed before moving to acceptance:

References: There are a large number of references that are older than 10 years. Surely, you could update and balance this list with more recent literature in the last few years, (2020 - 2025). 

Methods: You need to clarify inclusion and exclusion criteria and provide more detail in the selection process.

Results/Discussion: You do a good job of discussing the psychometric properties, but could use to better synthesize and expand on the limitations including the heterogeneity of the studies.

Language: Although I think, this is a minor issue, some of your sentences run a little long and assumed repeated; some editing for concise language and clarity would make for easier reading.

Over all, the paper includes a timely and pertinent review, and I can see the potential for contribution to the knowledge base, but a few revisions are required to improve the rigor and alignment with journal requirements.

Comments on the Quality of English Language

The language used is mostly clear and easy to understand, but some sentences tend to be lengthy and repetitive. A more concise approach would enhance clarity.

Author Response

(The authors gave the same response as above.)
